# Exploring multiple block-based 3D volumes utilising two-handed multiple degrees-of-freedom interaction

## ABSTRACT

The primary purpose of this paper is to run isotonic and isometric experiments on a multiple degrees-of-freedom input and two-handed interactive framework for the analysis of computer-aided visualisations. The key novelty of the system allows different multi-degree-of-freedom devices to be interacted simultaneously, allowing more intuitive and natural behaviour. In the evaluation, we performed the virtual removal and extraction of a palaeontological 3D model from its host rock using rotation and translation of subvolumes of data followed by interactive 2D transfer volume visualisation manipulation. We use a "brick" definition to divide the geological data sets into manageable portions that can be combined interactively in a similar way that a person with two hands combines and creates structures with LEGO blocks. We discuss and present for this application its accuracy and completion time.

## CCS CONCEPTS

• **Human-centered computing → HCI design and evaluation methods**; **Usability testing**; **Interaction techniques**; **Visualization toolkits**.

## KEYWORDS

Linear function, non-linear function, transfer function, isotonic, isometric

**ACM Reference Format:**
. 2020. Exploring multiple block-based 3D volumes utilising two-handed multiple degrees-of-freedom interaction. In *Proceedings of ACM Conference (Conference'17)*. ACM, New York, NY, USA, 4 pages. https://doi.org/10.1145/nnnnnnn.nnnnnnn

## 1 INTRODUCTION

LEGO blocks are solid shapes used for construction play. Studies [6] [11] suggest that these toy blocks are powerful learning tools to develop motor skills and hand-eye coordination, spatial reasoning, and linked with higher mathematical achievement. Gaming is the most apparent application of LEGO blocks in 3D visualisations [9] [12]. However, 3D Block building systems are not limited for gaming but also applied in other areas of simulation and training, and scientific visualisation.

For example, in palaeontology, these computer-aided visualisation block-techniques allow fossils to be characterised in three dimensions in unprecedented detail empowering individuals and teams anywhere in the world to collaborate, visualise and interact with them. Thus, palaeontologists can construct and compare

*Conference'17, July 2017, Washington, DC, USA*
© 2020 Association for Computing Machinery.
ACM ISBN 978-x-xxxx-xxxx-x/YY/MM...$15.00
https://doi.org/10.1145/nnnnnnn.nnnnnnn

3D solid models to gain essential insights into the anatomy, development, and preservation of fossil organisms and apply different volume sculpturing techniques to, i.e. define high-resolution and cross-sectional views of body fossils from a variety of host rocks [4][15].

Several techniques exist for non-destructively characterising fossils in 3D, generating a series of parallel slices which map X-ray attenuation through the specimen. However, one concurrent problem when managing this vast amount of volumes in virtual environments (VEs) is misalignment, and users can utilise image registration and generation techniques as a way to fix this problem. In this paper, we aligned four volumes of an X-ray computed tomography dataset within Drishti [10], an open-source Volume Exploration and Presentation Tool, and Library for Interactive Settings of User-Mode (LISU) [16], a bimanual interaction software framework to transform individual components with one hand while adjusting other components for example transfer functions with the other hand simultaneously.

## 2 BIMANUAL BRICK MANIPULATION

For the purpose of this investigation, we define:

- **Linear functions** as the functions used to implement geometric operations such as rotation and translation
- **Non-linear functions** as the functions with discontinuities and/or a variable rate of change, such as lighting or filtering.
- **Transfer functions** as the non-linear functions in Drishti that define which voxels are visualised, based on the opacity absolute and gradient values collected with the original scan data to remove the noise and highlight features.

Transfer functions are represented by a 2D histogram panel that appears on the right-hand side of the window of Drishti (fig. 1).

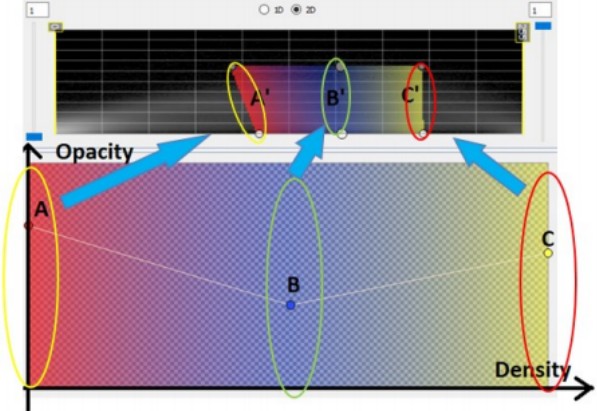

**Figure 1: Drishti's 2D histogram panel which allows users to change the colour and opacity of selected voxels.**

LISU's multilayered structure allows to apply **linear and non-linear functions simultaneously**, by using **multiple controllers** and combining the degrees-of-freedom (DOF).

By using Drishti, LISU and two high DOF different controllers, we reconstructed a *Panthera pardus* shown in figure 2 [13].

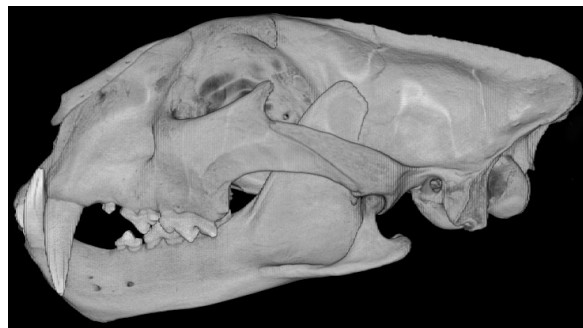

Figure 2: X-ray CT scan used in the present study.

## 2.1 Bricks

Drishti provides a sculpting facility by subdividing the volume data to explore into smaller "chunks" similar to LEGO blocks called *bricks*. Each brick can be textured with its own set of *transfer functions* and can be rotated, translated and scaled. For a brick's control and manipulation, we aimed at a manual asymmetrical two-handed technique, following recommendations from [8] [5], to manually move the brick objects to the desired direction resembling of playing a LEGO block videogame and using both hands for different functions.

## 2.2 Controllers

Controllers can be classified into two types, isotonic (displacement), for example the passive mouse or isometric (force), for example the joystick. Interactive manipulations of 3D visualisations require users to select 3D positions or 3D objects by combining controller input with these two types of interactions across the required number of DOF [14], enhancing users' experience [2].

For this study, we selected a **3Dconnexion SpaceNavigator** (fig. 3a) [1], and a **Wing** controller (fig. 3b) [18]. These controllers combined provide **bimanual input**, up to **12DOF**, with isotonic thumb joysticks and several buttons which satisfy the requirements for this study. Buttons are used to change actions in LISU, from the rotation to translations in VE, and analogue sticks are used to set the input value of a parameter, normalized from -1 to 1.

## 2.3 Image registration process

In the present study, we loaded four volumes with four different transfer functions, representing 477 slices, each slice 0.5 mm thick with an interslice spacing of 0.5 mm. of an X-ray CT scan of a *Panthera pardus* head taken along the coronal axis, running ten trials with the pair of controllers of figure 3. The operator was right-handed and figure 4a shows the resulted four volumes loaded in Drishti.

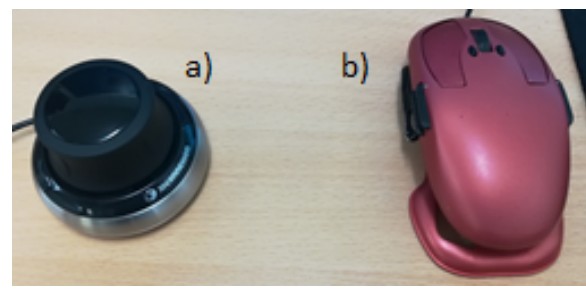

Figure 3: Controllers selected with isometric and isotonic properties for bimanual brick manipulation.

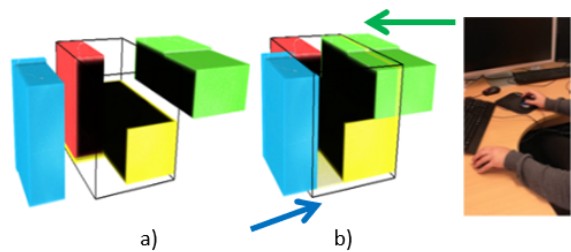

Figure 4: a) Four volumes of a X-ray computed tomography dataset are loaded into Drishti. Each volume is represented by a brick and has a related transfer function. b) The operator is tasked to translate and rotate the individual bricks to the voxel area.

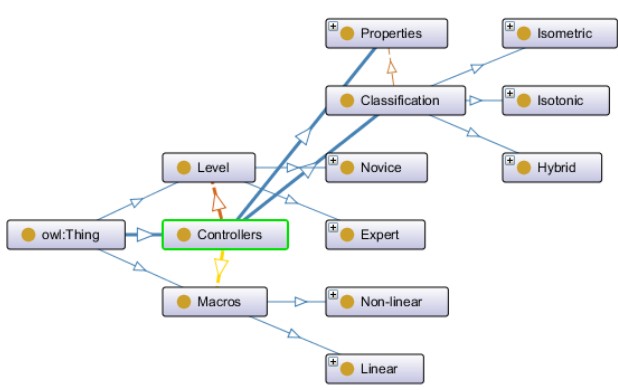

Figure 5: Input Devices' Ontology applied to direct functions to the controllers.

*2.3.1 Moving the bricks to the voxel area.* $V_{reference} : \mathbb{R}^3 \mapsto \mathbb{R}$ is the reference volume, the volume conformed by the yellow and red bricks in fig. 4a. The green brick, denoted as $V_{f_{green}} : \mathbb{R}^3 \mapsto \mathbb{R}$, and the blue brick, denoted as $V_{f_{blue}} : \mathbb{R}^3 \mapsto \mathbb{R}$, are the floating volumes to be registered to $V_{reference}$. The goal of this 3D rigid-body image registration is to estimate the rigid-body transformation $T_{V_1} : \mathbb{R}^3 \mapsto \mathbb{R}$ and $T_{V_2} : \mathbb{R}^3 \mapsto \mathbb{R}$ for which the transformed floating bricks $V_{LISU_1} \circ V_{f_{green}}$ and $V_{LISU_2} \circ V_{f_{blue}}$ are aligned

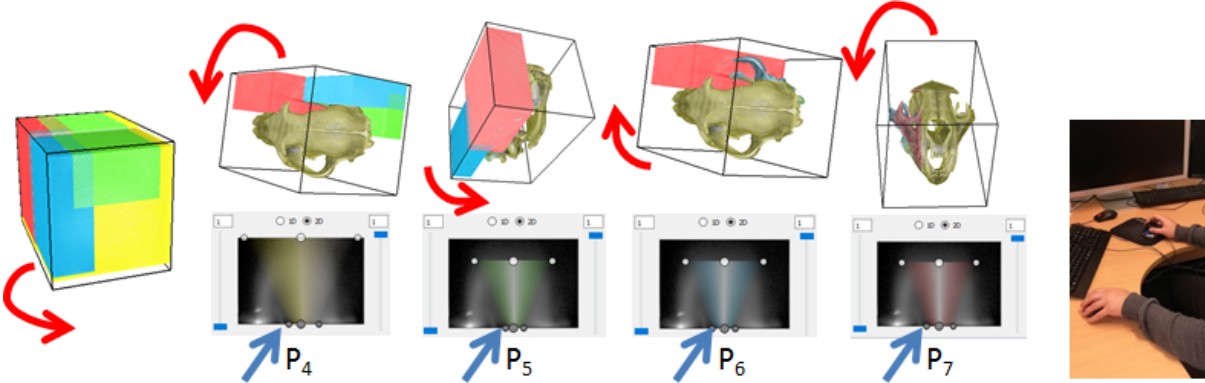

**Figure 6: The operator applies a transfer function to each brick to remove layers of density (the outer shell of the reconstructed image), indicated with the blue arrow, and rotates the camera view to see the inner structure, indicated with the red arrow.**

with $V_{reference}$. $V_{reference}$ dimensions were $1155 \times 1166 \times 1809$ millimeters (mm), the blue brick dimensions were $1023 \times 392 \times 1809$ mm, and the green brick dimensions were $446 \times 978 \times 1809$ mm.

According to [7], $T_{V_1}$ and $T_{V_2}$ can be represented by $4 \times 4$ homogeneous transformation matrices, such that a point $\overrightarrow{p} = [x, y, z, 1]$ is transformed to $T\overrightarrow{p}$. Then, $T_{V_1}$ is parameterized with 3 translations $[t_x, t_y, t_z]$ and 3 rotations $[\theta_x, \theta_y, \theta_z]$ such as $T_{V_1}(t_{x_1}, t_{y_1}, t_{z_1}, \theta_{x_1}, \theta_{y_1}, \theta_{z_1})$ and $T_{V_2}$ is $T_{V_2}(t_{x_2}, t_{y_2}, t_{z_2}, \theta_{x_2}, \theta_{y_2}, \theta_{z_2})$, where input values are normalized. We define $P_1$ as the macro that applies $T_{V_1}$ and $P_2$ as the one that applies $T_{V_2}$. Any $P$ process is composed out of the composition of resulting processes directed by the LISU's **ontology layer** in a **algorithm layer (AL) matrix**. LISU's AL matrix combines these macros into a final process $P_{final}$ such as: $P_{final} = P_1 P_2 = \prod_{i=1}^{2} P_i$.

Figure 4b shows how $P_{final}$ translates the blue and green bricks simultaneously. For the matching process of volumes, we define the effectiveness of registration (or how well the volumes are aligned) when **the four volumes meet inside the voxel cube**. We achieved this by stacking the bricks together, similar to play with LEGO blocks and implementing a $P_3$ macro to rotate the camera and explore the new volume ($P_{final} = \prod_{i=1}^{3} P_i$).

*2.3.2 Changes to the ontological component.* [17] proposes an ontology-based on the nature of the controller. However, this study implemented the ontology in figure 5. These changes to the ontological version in[17] give the result a more robust ontology to direct processes based on the number of DOF of the system and the level of expertise of users. From this ontology, a term is inductively defined by two rules, being: 1) *every variable and constant is a term*, and 2) *if f is a n-ary function and $t_1, ..., t_n$ are terms, then $f(t_1, ..., t_n)$ is also a term*. The combined system allows 12DOF, so applying this rule, $P(f_1, ..., f_{12})$, where rotations and translations can be interchangeable. Then, the ontology component directs $P_1$ to the right controller (3Dconnexion SpaceNavigator) and directs $P_2$ to the left controller (the Wing), a decision made by the rules and constraint of the controllers detected by the ontology that changes the actions that entities can perform: $\forall x, y(translates(x, y) \rightarrow controller(x) \wedge level(y))$ or $\forall x, y(rotates(x, y) \rightarrow controller(x) \wedge level(y))$.

For this study, both controllers have large DOF systems (combined is 12 DOF), so the operator can choose the predilected controller to perform each action. In the case of there being fewer DOF, based on the constraints of the controllers, the operator through LISU can select the operations with the most appropriate controller. The aim is for the system to be able to select the best option but for the operator to remain in control. In this experiment, the LISU ontology directed $P_3$ to the left controller for the rotation of the camera.

*2.3.3 Linear and non-linear functions applied simultaneously.* We define $P_4$, $P_5$, $P_6$, and $P_7$ as the macros to modify the transfer function for each brick, to expose the interior of the substantial region by slicing along significant axes. Any coordinate $(x, y)$ on the Drishti's histogram represents the set of points with the density $x$ and gradient of density $y$. LISU's combined 12DOF allows the user to modify the coordinates of the histogram with both controllers, selecting a different brick transfer function via the buttons of the controllers. $P_{final}$ is again modified and is in the form of: $P_{final} = \prod_{i=1}^{7} P_i$. Figure 6 shows how $P_4$ to $P_7$ map the colour and opacity to the voxels while the operator rotates the camera.

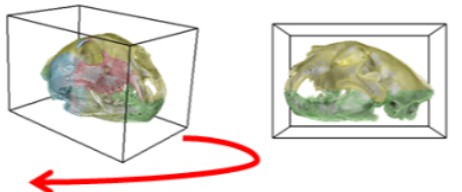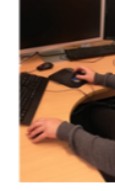

**Figure 7: After a final exploration and comparison to the original image, the operator decides if the reconstruction was successful or not.**

*2.3.4 Visual evaluation of image differences.* Figure 7 shows the operator using $P_3$ to rotate the camera to find any possible fault. The displacement of the two bricks placed the bricks inside the voxel cube, and the RMS error between corresponding pixels after

registration was 0, having a perfect reconstruction. After a final exploration, the reconstruction is considered exact as it meets the parameters of the original image, finalising the image reconstruction process.

## 3   RESULTS AND CONCLUSIONS

All movements were tracked and recorded. Figure 8 show a trend of using the dominant hand for more accurate and precise movements, whereas the non-dominant side only performed slight movements. The green brick *distance* was 765.80 mm, moving from left to right. The blue brick *distance* was 1008.07 mm, moving from right to left. Each movement trajectory had a duration of 0.58 s and consisted of a Gaussian velocity profile. The translation amplitude was 1173.87 mm (total displacement), with a peak acceleration of 1.0215 $mm/s^2$ and a peak velocity of 41.3451 $mm/s$. The completion time was $F(1, 10) = 45.57344$ $s$ (ANOVA, $p < 0.05$), and it can be reduced, depending on the CPU capability, ability of the operator and familiarity with the input devices.

Results show that LISU is an efficient tool to work with linear and non-linear functions simultaneously, in this case, to work with bricks movement and Drishti's transfer functions. LISU provided full control over the virtual object via its ontological component and AL matrix, gaining an understanding of the material, and natural and immersive experience for users. Further research is needed to evaluate LISU performance and we are planning to develop more complex experiments involving higher DOF and more controllers such as gamepads and hand gestures systems as bimanual input can substantially improve human-computer interaction [3]. A more extensive survey and evaluation are planned using more bricks, aiming different disciplines, such as geology, to study rock cuttings from oil and gas wells to locate and collect useful hydrocarbons. The result is the image registration or alignment of multiple layer-cake volumes representing geological phenomena where each layer describes a type of rock material and property.

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

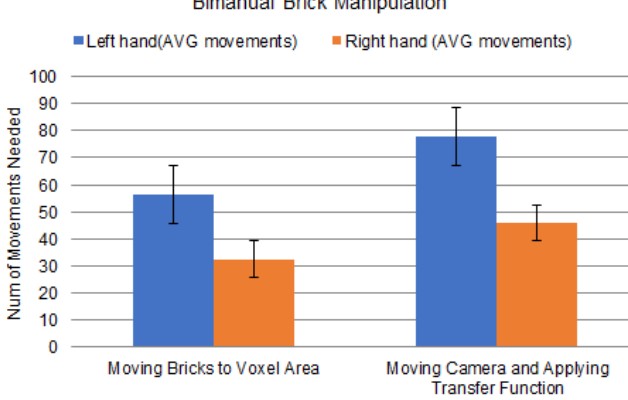

**Figure 8: Number of movements executed by hand to complete the image registration task. Right hand is the dominant hand and left hand is the non-dominant side.**