# OpenReview forum: "Exploring multiple block-based 3D volumes utilising two-handed multiple degrees-of-freedom interaction"
_graphicsinterface.org/Graphics_Interface/2020/Conference — Submitted to GI 2020_

### Official Review · AnonReviewer1 · 2020-02-04

**Rating:** 3
**Confidence:** 4

**Review:**

This paper describes the use of a bimanual interaction software framework and two input devices to perform a task in which volumes of 3D data representing a CT scans of a fossil are re-aligned and the transfer function for visualizing the voxels is manipulated.

I find it difficult to evaluate this work because it does not clearly identify the contributions that it makes to graphics, HCI, or visualization research, and it does not present a structured review of related work.

The abstract states that the "primary purpose of this paper is to run isotonic and isometric experiments on a multiple degrees-of-freedom input and two-handed interactive framework", and that the key novelty of the system is that it "allows different multi-degree-of-freedom devices to be interacted simultaneously". But input systems in which users manipulate 3D data with more than one multiple-DOF controller are commonplace (e.g., current commercial VR systems), and textbooks have been written on human input that develop formal definitions of devices based on the DOF they enable (e.g., Bill Buxton's online text on the topic https://www.billbuxton.com/inputManuscript.html).

I am not an expert on the specific application area that this paper looks at (that of working with CT scan data), but given the maturity of research on multi-DOF interaction techniques in the broader HCI literature, I think much more is needed to justify the novelty of this work.

Beyond issues of contribution and framing, the presented study appears to have been conducted with only one participant, so it at best can be considered as a case study. As well, the methodology is not described in anything close to a rigorous or reproducible manner.

Considering all of the above, my recommendation is that this submission not be accepted.

---

### Official Review · AnonReviewer2 · 2020-02-09
**Lacking a Lot**

**Rating:** 3
**Confidence:** 4

**Review:**

The paper mainly describes conducting a palaeontology experiment of using a system to manipulate 3D data using bimanual 12-DOF control to align blocks of CT data. The purpose of the paper is to describe this experiment. There is no description of prior methods to compare to. There is not enough evaluation on how well the example performs. There is no novelty in the paper. Although the authors suggest some novelty in the 12-DOF interaction system. But there is no comparison of prior work to prove it is novel. If the authors' intention is to introduce this system, they should also use more experiments to show the advantage of the system. It appears the authors merely use existing software (Drishti and LISU) to build the system. And there is no reference to LISU in [16], no conference, journal or webpage.

If the authors want to revise the paper to focus on the novelty of the system, they should highlight the key component that is novel, list prior work of similar systems, analyze the differences and conduct more experiments to prove their system works better generally. If the authors want to focus on user study instead, they should conduct more experiments and give more detailed analysis on how the system helps users under different usage scenarios, different tasks and different system settings.

---

### Official Review · AnonReviewer3 · 2020-02-17
**Rejected**

**Rating:** 3
**Confidence:** 3

**Review:**

This manuscript introduces a visualization system that allows bimanual manipulation of the virtual content. The system is evaluated using the virtual removal and extraction tasks. Unfortunately, this manuscript is poorly written that I am not sure if this manuscript wants to introduce a novel system for bimanual manipulation or justify a bimanual system is suitable for interactive visualization tasks.

-	From the manuscript, I cannot see any novelty regarding to the visualization system.
-	If this manuscript is to justify the system is suitable for interactive visualization, then I suggest that it should have a target hypothesis to be justified in the very beginning. Then the authors need to introduce the experiment design which may include the system, experimental protocol and tasks for subjects to finish. But little has been furnished in current manuscript and the results are limited to comparison of the needs of manipulation for different hands.
-	While in the abstract the authors mention virtual removal and extraction tasks, it seems little has been discussed about the two tasks, but some assembly tasks and lighting adjustment are involved.
-	I would also like to know how 12DOF is supported by a joystick and a mouse. Would it be too counterintuitive so that the needs for operations may increase?

I therefore vote reject and hope the authors would spend large effort to improve their manuscript.

---

### Decision · Program_Chairs · 2020-02-18

Reject